# A Molecular Signature Determines the Prognostic and Therapeutic Subtype of Non-Muscle-Invasive Bladder Cancer Responsive to Intravesical Bacillus Calmette-Guérin Therapy

**DOI:** 10.3390/ijms22031450

**Published:** 2021-02-01

**Authors:** Seon-Kyu Kim, Seong-Hwan Park, Yeong Uk Kim, Young Joon Byun, Xuan-Mei Piao, Pildu Jeong, Kyeong Kim, Hee Youn Lee, Sung Pil Seo, Ho Won Kang, Won Tae Kim, Yong-June Kim, Sang-Cheol Lee, Sung-Kwon Moon, Yung Hyun Choi, Wun-Jae Kim, Seon-Young Kim, Seok Joong Yun

**Affiliations:** 1Personalized Genomic Medicine Research Center, Korea Research Institute of Bioscience and Biotechnology (KRIBB), Daejeon 34141, Korea; seonkyu@kribb.re.kr (S.-K.K.); sungghwan@kribb.re.kr (S.-H.P.); kimsy@kribb.re.kr (S.-Y.K.); 2Department of Bioinformatics, KRIBB School of Bioscience, Korea University of Science and Technology, Daejeon 34113, Korea; 3Department of Urology, College of Medicine, Yeungnam University, Daegu 38541, Korea; jojo9174@daum.net; 4Department of Urology, Chungbuk National University College of Medicine, Cheongju 28644, Korea; lenic0819@naver.com (Y.J.B.); phm1013@hotmail.com (X.-M.P.); leo24fly@hanmail.net (P.J.); uro_kk@naver.com (K.K.); leeheeyoun1@hanmail.net (H.Y.L.); spseo79@gmail.com (S.P.S.); howon98@naver.com (H.W.K.); wtkimuro@chungbuk.ac.kr (W.T.K.); urokyj@cbnu.ac.kr (Y.-J.K.); lscuro@chungbuk.ac.kr (S.-C.L.); wjkim@chungbuk.ac.kr (W.-J.K.); 5Department of Urology, Chungbuk National University Hospital, Cheongju 28644, Korea; 6Department of Food and Nutrition, College of Biotechnology & Natural Resource, Chung-Ang University, Anseong 06974, Korea; sumoon66@cau.ac.kr; 7Department of Urology, Personalized Tumor Engineering Research Center, Chungbuk National University College of Medicine, Cheongju 28644, Korea; choiyh@deu.ac.kr; 8Korea Bioinformation Center, Korea Research Institute of Bioscience and Biotechnology (KRIBB), Daejeon 34141, Korea

**Keywords:** non-muscle-invasive bladder cancer, molecular subtypes, gene signature, prognosis, BCG

## Abstract

Non-muscle-invasive bladder cancer (NMIBC) is clinically heterogeneous; thus, many patients fail to respond to treatment and relapse. Here, we identified a molecular signature that is both prognostic and predictive for NMIBC heterogeneity and responses to Bacillus Calmette-Guérin (BCG) therapy. Transcriptomic profiling of 948 NMIBC patients identified a signature-based subtype predictor, MSP888, along with three distinct molecular subtypes: DP.BCG+ (related to progression and response to BCG treatment), REC.BCG+ (related to recurrence and response to BCG treatment), and EP (equivocal prognosis). Patients with the DP.BCG+ subtype showed worse progression-free survival but responded to BCG treatment, whereas those with the REC.BCG+ subtype showed worse recurrence-free survival but responded to BCG treatment. Multivariate analyses revealed that MSP888 showed independent clinical utility for predicting NMIBC prognosis (each *p* = 0.001 for progression and recurrence, respectively). Comparative analysis of this classifier and previously established molecular subtypes (i.e., Lund taxonomy and UROMOL class) revealed that a great proportion of patients were similar between subtypes; however, the MSP888 predictor better differentiated biological activity or responsiveness to BCG treatment. Our data increase our understanding of the mechanisms underlying the poor prognosis of NMIBC and the effectiveness of BCG therapy, which should improve clinical practice and complement other diagnostic tools.

## 1. Introduction

Bladder cancer is responsible for 150,000 deaths annually in the world [1]. Non-muscle-invasive bladder cancer (NMIBC) is the most common subtype of this disease, accounting for approximately 80% of all cases. Approximately 20% of these patients experience disease progression to muscle-invasive bladder cancer (MIBC) after treatment, a development associated with a very poor prognosis. Conventional histopathological parameters, including stage, grade, tumor size, number of tumors, and presence of concomitant carcinoma in situ (CIS), are generally considered to be prognostic factors. However, the utility of these factors for precise prediction of patient outcomes is limited [2]. Bacillus Calmette-Guérin (BCG)-based immunotherapy combined with transurethral resection (TUR) is the optimal treatment for preventing or delaying recurrence and/or progression of NMIBC [2,3]. However, many patients, especially those with T1 high-grade tumors, fail to respond to BCG therapy and are at a high risk of recurrence and progression [4,5,6].

Much research has been conducted to better understand tumor heterogeneity at a molecular level and to establish prognostic models of NMIBC [7,8,9,10,11]. Although most studies of bladder cancer focused primarily on MIBC [12,13,14], or on combined analysis of NMIBC and MIBC [15], recent studies have examined NMIBC exhibiting several distinct molecular subtypes in an attempt to address tumor heterogeneity [7] and responsiveness to BCG therapy [8]. Briefly, Sjödahl et al. [15] first reported five molecular subtypes of bladder cancer (i.e., Urobasal A, Genomic Unstable (GU), Infiltrated, Urobasal B, and Squamous cell carcinoma (SCC)-like in Lund Taxonomy), which partially addressing NMIBC subtypes. The European UROMOL consortium [7] performed transcriptomic profiling using more than 400 patients with NMIBC, and uncovered three molecular subtypes of NMIBC (i.e., Classes 1~3 in UROMOL classes). More recently, Robertson et al. [8] also reported the five consensus subtypes of T1 tumors (i.e., T1-Myc, T1-LumGU, T1-Inflam, T1-TLum, and T1-Early) in the patients who were treated with TUR and BCG treatments. Although previous studies identified the molecular characteristics of NMIBC, they did not provide enough information to identify patients most likely to respond to BCG treatment, or they did not validate the therapeutic relevance independently. Thus, to increase the utility of BCG therapy, further rigorous exploration of the molecular environment of NMIBC, along with identification of new biomarkers to re-stratify molecular subtypes of NMIBC, will facilitate assessment of prognosis and identify those most likely to benefit from BCG therapy.

Here, we investigated putative genetic signatures expressed by distinct molecular subtypes of NMIBC and examined their utility for predicting prognosis and responses to BCG treatment. Three subtypes of NMIBC exhibited distinct prognostic features: (1) DP.BCG+, associated with progression and positively responsive to BCG, (2) REC.BCG+, associated with worse recurrence-free survival and better responsive to BCG, and (3) EP, exhibiting equivocal prognostic behavior. To validate the utility of the signature, we constructed a signature-based classifier using a deep learning method and confirmed the independent clinical utility of the classifier and its therapeutic relevance to different subtypes in multiple independent patient cohorts.

## 2. Results

### 2.1. Establishment of Distinct Molecular Subtypes Associated with Prognosis of NMIBC

Appendix A details the baseline characteristics of the 948 NMIBC patients. Of the four patient cohorts, we first performed gene expression profiling of the Chungbuk National University (CBNU) cohort containing 103 patients with NMIBC. Unsupervised consensus clustering [16] of 1623 genes showing varying expression across NMIBC samples (standard deviation (SD) > 0.9) identified three major sample clusters that were clearly molecularly distinguishable among patients with NMIBC (Figure 1A,B), and these data are consistent with those in a previous report [7]. Considering the silhouette width of each NMIBC sample, five with negative silhouette values were discarded; the remaining 97 samples, whose gene expression patterns agreed with their respective sample clusters, were used to estimate prognosis (Figure 1C). Based on consensus clustering, three sample subgroups were significantly predictive of progression-free survival (PFS) in patients with NMIBC (log-rank test, *p* = 0.04; Figure 1D), demonstrating the prognostic relevance of the newly identified molecular subgroups. Using recurrence data for NMIBC patients in the CBNU cohort, we examined the relationships between each of the three subgroups and recurrence-free survival (RFS). However, we found no significant difference between clusters (Figure 1E), and this may be because the sample size was too small to analyze heterogeneous clinical events such as disease recurrence.

### 2.2. One Subtype, Cluster 3, Is Predictive of Disease Progression and a Better Response to BCG Treatment

Based on the molecular subgroups identified within the CBNU cohort, we further explored the prognostic characteristics of these subgroups in other independent NMIBC cohorts. Before classifying the patients in the other cohorts, we sought to identify optimal gene sets that distinguished the three NMIBC clusters in the CBNU cohort. First, we selected three lists of genes that showed significant differential expression between Cluster 1 and others (2990 genes), Cluster 2 and others (1201 genes), or Cluster 3 and others (1756 genes) (two-sample *t*-tests, *p* < 0.001; Figure 2A). Considering the fold differences in expression, we chose the top 400 genes (i.e., 200 upregulated and 200 downregulated genes in each sample cluster) from each gene list (Figure 2A). By aggregating these cluster-specific top-ranked genes (1200 genes in total (400 from each sample cluster)), we obtained a final gene signature comprising 888 genes, which we used to classify risk subgroups of NMIBC.

Using this signature, we applied a deep learning method based on a deep belief network (DBN) [17] to test the accuracy of our gene set-based prediction of disease progression (Figure 2B). The expression data for 888 genes in the CBNU cohort (the training set) were pooled to form a classifier able to estimate the probability that a particular NMIBC sample belonged to one of the three subgroups. To prevent overfitting the classifier, the prediction model was pre-trained using an auto-encoder algorithm [17]. When this molecular subtype prediction model defined by 888 genes (named MSP888) was applied to the UROMOL cohort (the test set), the classifier identified significant differences in the risk of NMIBC progression among the patients in the three subgroups (log-rank test, *p* = 1.93 × 10^−6^; Figure 2C). The progression rate in Cluster 3 was significantly higher than that in the other clusters (Figure 2C), a finding consistent with the previous result for the CBNU cohort (Figure 1D).

Adjuvant intravesical therapy with BCG is the best treatment option for handling disease events in those with high-risk NMIBC [2]. Because BCG treatment data for the UROMOL cohort were available, we sought to determine whether the MSP888 classifier predicted a potential benefit from BCG treatment. To assess the predictive value of the classifier in the UROMOL cohort, we stratified patients with NMIBC into three subgroups (i.e., Cluster 1, 2, or 3) and estimated differences in disease progression in each group independently. Notably, we confirmed that BCG treatment affected PFS significantly in patients in Cluster 3 (log-rank test, *p* = 0.001; Figure 2F), whereas there was no significant association between BCG treatment and progression in the other patient subgroups (log-rank test, *p* > 0.05 for each; Figure 2D,E).

To determine the prognostic independence of the signature-based classifier for NMIBC progression, we applied Cox regression analyses to the three subgroups in the UROMOL cohort, along with known clinicopathological risk factors. Univariate analysis identified stage and grade as significant prognostic indicators of PFS in NMIBC (Table 1). Multivariate analysis revealed that the MSP888 classifier retained statistical significance for the PFS of NMIBC patients (hazard ratio (HR), 3.447; 95% confidence interval (CI), 1.483–8.012; *p* = 0.004; Table 1), illustrating the high prognostic relevance of the three subtypes classified by the MSP888 classifier. Based on the characteristics of Cluster 3, we referred to it as the DP.BCG+ subtype, indicating a subtype associated with a poor prognosis in disease progression and a good response to BCG treatment.

### 2.3. One Subtype, Cluster 2, Is Associated with Disease Recurrence and a Better Response to BCG Treatment

Because another NMIBC patient cohort (the European cohort) provided NMIBC recurrence data, we also used the MSP888 classifier for that cohort to test prediction accuracy for disease recurrence (Figure 3A). We applied the prediction model built for the CBNU cohort to the European cohort and found significant differences in risk of NMIBC recurrence among patients in the three subgroups (log-rank test, *p* = 0.003; Figure 3B). Cluster 2 exhibited a worse prognosis than the other clusters (Figure 3B).

Also, we asked whether the classifier predicts the potential benefits of BCG treatment in the European cohort. We stratified patients with NMIBC into three subgroups (i.e., Clusters 1, 2, or 3) and estimated differences in disease recurrence in each group independently. Importantly, we observed that BCG treatment significantly affected RFS in patients in Cluster 2 (log-rank test, *p* = 0.005; Figure 3D). However, there was no significant association between BCG treatment and prognosis in the other patient subgroups (log-rank test, *p* > 0.05 for each; Figure 3C–E).

The prognostic independence of the classifier for NMIBC recurrence was assessed by applying Cox regression analyses to the three subgroups in the European cohort. Univariate analysis identified stage, grade, BCG treatment, and patient cluster as significant indicators of NMIBC recurrence (Table 1). Multivariate analysis showed that the MSP888 classifier retained statistical significance for RFS in NMIBC patients (HR, 2.569; 95% CI, 1.065–6.915; *p* = 0.036; Table 1), illustrating the high prognostic relevance of the subtype classified by the MSP888 predictor as an independent risk factor for NMIBC recurrence. Based on the characteristics of Cluster 2, we referred to it as REC.BCG+, indicating a subtype associated with a poor prognosis in disease recurrence and a better response to BCG treatment. We referred to Cluster 1 as EP, indicating a subtype showing equivocal prognosis.

### 2.4. Validation of Prognostic Subtypes in an Independent Cohort

To further verify the characteristics of the three subtypes (i.e., EP, REC.BCG+, and DP.BCG+) classified by our classifier, we constructed a transcriptomic dataset by conducting RNA sequencing (RNAseq) experiments in an independent NMIBC patient cohort (the CBNU-RNAseq cohort, *n* = 32). This cohort contained NMIBC patients in high-risk groups (any of the following: T1, high-grade, multiple, or large tumors (>3 cm)) that showed different clinical outcomes. The patients were stratified into three prognostic subgroups: Group 1 (patients with non-recurrence or non-progression, *n* = 15), Group 2 (patients experiencing recurrence two or more times, *n* = 9), and Group 3 (patients experiencing progression to MIBC, *n* = 8). The prognostic course of each patient group was confirmed clinically, with a long follow-up time (median, 75.4 months). The distribution of European Organization for Research and Treatment of Cancer (EORTC) risk scores among these subgroups is described in Appendix A, and there are non-significant differences in disease risk among groups. By applying the same procedure (i.e., a deep learning method), we built a prediction model for the CBNU cohort and applied it to the CBNU-RNAseq cohort (Appendix A). When the prediction model was applied to the CBNU-RNAseq cohort, the MSP888 classifier revealed significant concordance rates between clinically confirmed subgroups (Groups 1, 2, or 3) and molecular subtype (EP, REC.BCG+, or DP.BCG+; χ^2^ test, *p* = 3.28 × 10^−8^; Appendix A), demonstrating the prognostic relevance of these molecular subtypes.

### 2.5. Biologic Characteristics of the Molecular Subtypes of NMIBC

Based on the results demonstrating prognostic and predictive potential, we tried to explore the biologic characteristics of the three molecular subtypes. Gene set enrichment analysis of the 888 genes in the classifier was performed using Ingenuity Pathway Analysis (IPA) software. When searching for previously established pathways, we found that genes involved in many canonical pathways were significantly enriched (Appendix A). Among these, some were activated or inhibited common in two molecular subtypes, whereas others were specific to a particular subtype. The relationship among the significant pathways is displayed in Figure 4, in which two pathways are interconnected if they share molecules.

When exploring EP subtype-specific pathways, we found that genes involved in PD-1/PD-L1 cancer immunotherapy, DNA damage responses, Sirtuin signaling, and senescence pathways were plentiful (Appendix A). Interestingly, many genes associated with PD-1/PD-L1 cancer immunotherapy (i.e., *B2M*, *HLA-B*, *HLA-DPA1*, *HLA-DQA1*, and *HLA-DRA*) were significantly downregulated, whereas *FANCB* (involved in the DNA damage response) was significantly upregulated, suggesting that PD-1/PD-L1 inhibitors may be a treatment option for NMIBC patients with the EP subtype. Expression of genes in EP-specific pathways was validated in an independent cohort (the UROMOL cohort; Figure 5A).

We also explored REC.BCG+-specific pathways and observed that genes involved in Th1, EIF2 signaling, Wnt/β-catenin signaling, or osteoarthritis pathways were significantly enriched (Appendix A). The vast majority of genes involved in these pathways (i.e., *NOTCH1*, *CCND1*, *CD44*, *MMP7*, *WNT10A*, *ITGA3*, *TGFBR2*, and *FGFR3*) are implicated in regulation of cell proliferation, supporting the observation of frequent disease relapse in patients with the REC.BCG+ subtype. We also found that *FGFR3*, a member of the bladder cancer signaling pathway, was significantly downregulated, indicating an aggressive characteristic and a worse prognosis. In the aspect of *FGFR3* expression, those expression levels in the EP subtype were significantly higher than in the others across NMIBC patient cohorts (Appendix A), supporting a poor responsiveness to BCG treatment in the EP subtype and consistent with the previous investigation [18]. The alteration in expression levels of genes involved in REC.BCG+-associated pathways was validated in the UROMOL cohort (Figure 5A).

The DP.BCG+ subgroup demonstrated specific alterations in expression of genes involved in cell cycle-associated pathways, such as cell cycle control of chromosomal replication or cyclins/cell cycle regulation (Appendix A). Among them, many (i.e., *FOXM1*, *TOP2A*, *CCNB1*, *CCNB2*, and *CDC25A*) participate in DNA repair, supporting the worse prognosis observed for the DP.BCG+ subtype. Expression of genes involved in DP.BCG+-specific pathways was validated in the UROMOL cohort (Figure 5A).

When identifying active or inactive pathways common to three subtypes, we found that cell cycle-G2/M DNA damage checkpoint regulation and xenobiotic metabolism signaling pathways were common to the EP and REC.BCG+ subtypes (Figure 4). The REC.BCG+ and DP.BCG+ subtypes shared several pathways, including oxidative phosphorylation, glutathione redox reactions I, and glutathione-mediated detoxification (Figure 4). No pathways were shared between the EP and DP.BCG+ subtypes.

### 2.6. Comparison of Previously Known Molecular Subtypes with the MSP888 Predictor of NMIBC

Next, we used the UROMOL cohort to carry out a comparative analysis between well-established pathological or molecular subtypes and our classifier (Figure 5B,C). With respect to stage and grade, we observed higher percentages of high-grade T1 tumors in patients with the DP.BCG+ subtype (59 out of 164; 36%) than in those with the other subtypes (14 out of 166 (8.4%) with EP and five out of 130 (3.8%) with REC.BCG+; Figure 5B), demonstrating that most high-risk NMIBC patients had the DP.BCG+ subtype. Regarding CIS patients, we observed that a higher percentage had the DP.BCG+ subtype (42 out of 164 (25.6%) compared with 22 out of 166 (13.3%)) with EP and 7 out of 130 (5.4%) with REC.BCG+; Figure 5B). When estimating disease progression, not surprisingly, we found that the progression rate in those with DB.BCG+ (23 out of 164, 14%) was higher than that in those with the other subtypes (eight out of 166 with EP (4.8%) and 0 out of 130 with REC.BCG+; Figure 5B), which is consistent with the previous results (Figure 2C). The percentage of patients in the DP.BCG+ group with a high EORTC risk score (103 out of 164, 62.8%) was higher than that in the other subgroups (53 out of 166 (31.9%) for EP and 18 out of 130 (13.8%) for REC.BCG+; Figure 5B), accounting for the poor prognosis of patients with DP.BCG+. We note that more patients with a high EORTC risk score had the EP subtype than the REC.BCG+ subtype. These results indicate that the MSP888 classifier is not perfectly compatible with the EORTC risk scoring system and should therefore be utilized independently in clinical practice.

When comparing the BASE47 signature with our classification system, we did not observe any association between Luminal or Basal-like characteristics and MPS888-based molecular subtype (Figure 5C). When comparing the Lund taxonomy with the MPS888, we found that the vast majority of GU patients were classified into the DP.BCG+ subtype (127 out of 162 patients with the GU, 78.4%), whereas many tumors with the infiltrated or urobasal subtype were classified into the REC.BCG+ and EP subgroups, respectively (Figure 5C). These results underscore the predictive potential of the MSP888 classifier for high-risk patients with NMIBC. Lastly, regarding the UROMOL classes, we found that most NMIBC patients with DP.BCG+ were UROMOL class 2 (151 out of 164 patients with DP.BCG+, 92%), which is the subgroup with the poorest prognosis. Many patients with UROMOL class 1 had the REC.BCG+ subtype, while most patients with UROMOL class 3 had the EP subtype (Figure 5C). Although the sub-classification systems based on the UROMOL [7] and our classifier share a great proportion of patients with each subtype, they do not share significantly enriched pathways (Figure 5A) or responsiveness to BCG treatment (Figure 2D–F).

## 3. Discussion

NMIBC is clinically heterogeneous [5]. Despite considerable effort [7,8,9,10,11], the clinical heterogeneity of NMIBC means that there is a great need to identify subtypes with biological and clinical relevance. Here, we used data from multiple NMIBC patient cohorts to carry out transcriptome profiling analyses. The results identified a gene signature for distinct prognostic subtypes of NMIBC (EP, REC.BCG+, and DP.BCG+). The signature-based MSP888 classifier showed significant prognostic relevance independent from other pathological factors. Our classifier also showed therapeutic relevance in that patients with the DP.BCG+ subtype benefited (i.e., reduced disease progression) from BCG therapy, while patients with the REC.BCG+ subtype responded to BCG treatment with respect to reduced disease recurrence. The overall characteristics of the three molecular subtypes are summarized in Appendix A.

Considerable effort has been devoted to uncovering the molecular characteristics and establishing prognostic models of NMIBC [7,8,9,10,11,19]. Recent transcriptomic profiling investigations revealed several distinct molecular subtypes relevant to tumor heterogeneity [7] and responsiveness of BCG therapy [8]. Despite these contributions, previous studies provided limited information with respect to identifying patients likely to respond best to BCG treatment, or did not have the therapeutic relevance validated independently. Therefore, the ability to predict prognosis and response to BCG therapy remains a major clinical challenge. Through gene expression profiling of multiple cohorts (948 patients) with NMIBC, we identified a signature comprising 888 genes and developed a MSP888 predictor for classifying prognostic subtypes of NMIBC. Patients with NMIBC classified as DP.BCG+ potentially benefit from BCG treatment with respect to disease progression, whereas those with the REC.BCG+ subtype are responsive with respect to disease recurrence. Although sub-classification systems based on the previous known molecular subtypes [7,15] share a great proportion of patients in each subtype with our own classifier (Figure 5C), the MSP888 classifier reflects the different biological characteristics of each subtype (Figure 5A) and different responses to treatment Figure 2D–F and Figure 3C–E) [7,8,15]. These data underscore the importance of the molecular subtype defined by the MSP888 classifier as a potential prognostic and predictive signature for NMIBC.

Several lines of evidence strongly support the signature-based MSP888 classifier as an independent and significant predictor of prognosis for NMIBC. First, the MSP888 classifier was a significant predictive factor for progression in the UROMOL cohort according to multivariate analysis (upper panel in Table 1). Second, the MSP888 classifier was an independent predictor of disease recurrence in patients with NMIBC in the European cohort, regardless of current histopathological criteria (lower panel in Table 1). Third, the MSP888 classifier was an independent predictor of disease events in patients with NMIBC, regardless of the current risk scoring system (Appendix A and Figure 5B). Taken together, these results suggest that the MSP888 classifier strongly retains its prognostic relevance, even after additional pathologic prognostic characteristics are taken into account.

The current study identifies several molecular characteristics and credible signaling pathways responsible for the three distinct subtypes. Regarding the EP subtype, PD-1/PD-L1 cancer immunotherapy and DNA damage responses are related to downregulation of *B2M* and *HLA* (*HLA-B*, *HLA-DPA1*, *HLA-DQA1*, and *HLA-DRA*) and to upregulation of *FANCB*. These results imply that PD-1/PD-L1 inhibitors may be a suitable treatment for NMIBC patients with the EP subtype. However, the effectiveness of PD-1/PD-L1 inhibitors against the EP subtype should be validated in other NMIBC cohorts since we had no treatment data relevant to PD-1/PD-L1 inhibitors in patients with NMIBC. The REC.BCG+ subtype was associated with a number of activated functions, including Th1, EIF2 signaling, and Wnt/β-catenin signaling pathways, supporting the aggressive characteristics of this subtype with respect to disease recurrence. Many genes activated in the REC.BCG+ subtype were associated with cell proliferation. Among these, *NOTCH1*, *CCND1*, *CD44*, *MMP7*, *TGFR2*, and *FGFR3* are involved in bladder cancer signaling pathways or are associated with the development of bladder carcinoma. In particular, *FGFR3*, a biomarker for prognosis of bladder cancer, was significantly inhibited in REC.BCG+ when compared with the other subtypes, supporting the aggressive characteristic of that subtype [20]. The DP.BCG+ subtype clearly showed strong alteration of cell cycle or DNA repair systems (Figure 5A). In this subtype, *FOXM1*, *TOP2A*, *CCNB1*, *CCNB2*, and *CDC25A* participate in DNA repair and are activated during disease relapse [9,10,11,20] or play a role in the prognosis of other cancers [21,22], thereby supporting the poor prognostic characteristics of the DP.BCG+ subtype.

In conclusion, we identified a gene signature comprising 888 genes and developed the MSP888 classifier to identify distinct prognostic subtypes of NMIBC. We also evaluated responses to BCG treatment and found that two subtypes (REC.BCG+ and DP.BCG+) show better responses to BCG immunotherapy. Taken together, these results increase our understanding of the genome-wide characteristics of NMIBC and supplement the prognostic or predictive characteristics identified in previously established molecular subtypes.

## 4. Materials and Methods

### 4.1. Public Datasets of NMIBC Patients

Three independent cohorts of patients (*n* = 916) with NMIBC, all of whom had publicly available gene expression data, were used for this study. Gene expression and clinical data from 165 primary bladder cancer patients were obtained from Chungbuk National University Hospital [23]. Among these, 103 NMIBC samples were used as the discovery cohort (the CBNU cohort, *n* = 103). Collection and analysis of samples were approved by the institutional review board of Chungbuk National University, and all subjects provided informed consent. Gene expression data from the CBNU cohort are available in the National Center for Biotechnology Information (NCBI) Gene Expression Omnibus (GEO) database under data series accession number GSE13507. An RNAseq dataset from the European UROMOL consortium [7] was downloaded from the ArrayExpress database under accession number E-MTAB-4321. In this dataset, 460 NMIBC samples were used as a validation cohort (the UROMOL cohort, *n* = 460). Another gene expression dataset from a European consortium for bladder cancer microarray study (GSE5479) was used for validation [24]. Among these, 353 NMIBC samples were selected as another validation dataset for the current study (the European cohort, *n* = 353). All transcriptomic data used in this study contain patient survival and follow-up time data to estimate the prognostic relevance of the signatures. The baseline characteristics of the NMIBC patient cohorts with publicly available datasets are described in Appendix A.

### 4.2. Patients and Tissue Samples of a Validation Cohort

To generate RNAseq data for patients in the CBNU-RNAseq cohort (*n* = 32), NMIBC patients who showed distinct clinical outcomes were selected (15 patients with any recurrence or progression, 9 who experience recurrence two or more times, and 8 showing MIBC progression), and all were in intermediate- to high-risk groups. Tumor tissues were collected from surgically resected NMIBC at Chungbuk National University Hospital, Chung-ju, Korea. All tumors were macro-dissected, typically within 15 min of surgical resection. Each NMIBC specimen was confirmed as representative by analysis of adjacent tissue in fresh frozen sections from TUR specimens, and then frozen in liquid nitrogen and stored at −80 °C until use. The collection and analysis of all samples in the CBNU-RNAseq cohort was approved by the Institutional Review Board of Chungbuk National University, and informed consent was obtained from each subject. The baseline characteristics of the CBNU-RNAseq cohort are described in Appendix A. In the current investigation, disease recurrence was defined as relapse of primary NMIBC at a lower or equivalent pathologic stage (Ta or T1). Disease progression was defined as an increase in stage from either Ta or T1 to T2 or higher after relapse [25].

### 4.3. RNA Extraction, RNAseq Experiments, and Data Processing

Total RNA was extracted from tissues using TRIzol reagent (Invitrogen, Carlsbad, CA, USA) and stored at −80 °C. The quality and integrity of the RNA were confirmed by the RNA Integrity Number (RIN) and the DV200 metric using an RNA 6000 Nano Kit and an Agilent 2100 Bioanalyzer (Agilent Technologies, Santa Clara, CA, USA). RNA samples with a RIN higher than five were designated as “good total RNA quality” and selected for downstream application. The sequencing library was prepared using TruSeq RNA Sample Preparation kit v2 (Illumina, San Diego, CA, USA). In brief, mRNA was purified from total RNA using poly-T oligo-attached magnetic beads, fragmented, and converted into cDNA. Adapters were then ligated to the cDNA, and the fragments were amplified by polymerase chain reaction (PCR). Sequencing of paired-end reads (2 × 150 bp) was performed using the HiSeq 2500 platform (Illumina, San Diego, CA, USA). Reference genome sequence data from *Homo sapiens* were obtained from the NCBI Genome database (assembly ID: GRCh38). Reference genome indexing and read mapping of tissue samples were performed using STAR software (ver. 2.5.4b). The dataset generated by RNAseq is available in the NCBI GEO public database under data series accession number GSE163899.

### 4.4. Transcriptomic Profiling

All gene expression data used in the current study were log2-transformed and quantile-normalized. The datasets from the CBNU or European cohorts contain gene expression values generated from commercial (Illumina human-6 v2.0 expression beadchip) or custom microarray (MDL human 3k oligo array) platforms, respectively. The fragments per kilobase of transcript per million fragments mapped value of each sample was measured in the UROMOL cohort, whereas the read counts per million fragments mapped value of each sample was measured in the CBNU-RNAseq cohort.

For unsupervised cluster analysis, 1623 genes showing varying expression across NMIBC samples (SD > 0.9 across 103 samples) in the CBNU cohort were selected. To determine the optimal number of possible sample clusters, consensus cluster analysis was applied to the gene expression data from 1623 genes, and the area under the consensus distribution function curve was compared across the clusters (*k* = 2–10). Consensus cluster analysis was carried out using the *ConsensusClusterPlus* [16] package in R language. Silhouette width was calculated to determine the accuracy of the clustering assignments. Only samples with a positive silhouette value were retained for downstream analysis because they best represented each subtype (R package: *cluster*).

### 4.5. Development of a Signature-Based Classifer

To validate the prognostic value of the molecular signature, a prediction model was generated by adopting a DBN [17] algorithm. Briefly, the model incorporated 888 genes that were differentially expressed among subtypes in the CBNU cohort according to two-sample t-tests. To construct a fully connected neural network, five hidden layers, in which 600, 300, 100, 300, and 600 nodes were allocated respectively, were set up. When training the model, a *Tanh* function was used as an activation function for the neural network, and a training procedure was repeated during 1000 epochs. To prevent overfitting or non-convergence of the classifier, the prediction model was pre-trained by an auto-encoder [17] algorithm. Development of the prediction model was undertaken using the H_2_O (https://www.h2o.ai) deep learning platform (ver. 3.32.0.2). A code snippet (in R language) for the current investigation is summarized as follows:


*# step of Auto-encoding*



*m1 <− h2o.deeplearning(1:888, training_frame = data_train, hidden = c(600,300,100,300,600),*



*auto-encoder = T, activation = “Tanh”, epochs = 1000)*



*# step of training*



*m2 <− h2o.deeplearning(x, y, training_frame = data_train, hidden = c(600,300,100,300,600),*



*pretrained_autoencoder = m1@model_id, activation = “Tanh”, epochs = 1000)*


### 4.6. Function Enrichment Analysis

Function enrichment analysis was carried out to identify the most significant canonical pathways associated with each molecular subtype. The significance of over-represented pathways was estimated by an overlap *p*-value and an activation *z*-score. The overlap *p*-value, estimated by Fisher’s exact test, measures whether there is statistically significant overlap between the genes in a dataset and the genes regulated by a pathway. The activation *z*-score is used to infer likely activation states of pathway candidates by comparison with a model that assigns a random regulation direction. A positive or negative activation *z*-score implies that a potential pathway is activated or inhibited, respectively. Function enrichment analysis was performed using the IPA tool (Qiagen, Valencia, CA, USA).

### 4.7. Other Statistical Analysis

To estimate the significance of differences in gene expression among the patient subgroups, a two-sample *t*-test (for two groups) or a one-way analysis of variance (ANOVA) test (for three groups) was performed, since gene expression datasets used in the current study revealed a normal distribution (Appendix A). Differences in expression were considered statistically significant if the *p*-value was <0.001. The Kaplan–Meier method was used to calculate RFS or PFS, and the difference in survival between two groups was assessed using log-rank statistics. The prognostic association between the classifier and known clinicopathological factors was assessed using multivariate Cox proportional hazard regression models. For comparison among categorical variables, Fisher’s exact test (two groups) or χ^2^ test (three groups) was performed. Statistical analysis was carried out in the R language environment (ver. 4.0.2).

### 4.8. Data Availability

The gene expression datasets from the CBNU, European, and CBNU-RNAseq cohorts are available in the NCBI GEO public database (https://www.ncbi.nlm.nih.gov/geo/) under data series accession numbers GSE13507, GSE5479, and GSE163899. The gene expression dataset from the European UROMOL consortium is available in the ArrayExpress database (https://www.ebi.ac.uk/arrayexpress/) under accession number E-MTAB-4321.

## Figures and Tables

**Figure 1 ijms-22-01450-f001:**
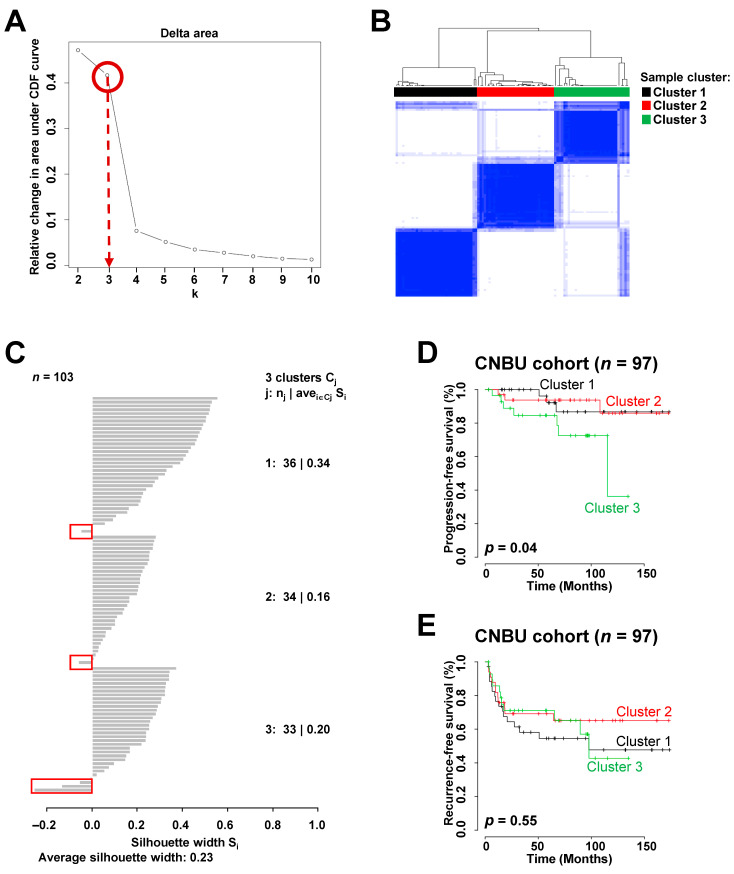
Identification of molecular subgroups and their association with prognosis in the Chungbuk National University (CBNU) cohort with non-muscle-invasive bladder cancer (NMIBC) (*n* = 103). A total of 1623 genes showing variation in expression across samples were selected for cluster analysis (standard deviation (SD) > 0.9). (**A**) Cluster stability estimated according to relative changes in the area under the consensus distribution function curve across k = 2–10, revealed that three clusters were clearly distinguishable in the NMIBC samples (the max slope of the curve occurs between three and four clusters). (**B**) Consensus matrix heatmap for k = 3. Data are presented in matrix format, in which white and blue denote no correlation (0) or a perfect correlation (1), respectively. (**C**) Silhouette widths of the CBNU cohort of 103 NMIBC samples. The figure shows the silhouette widths of samples in each cluster. Samples with positive Silhouette widths were selected to estimate the prognostic values of each sample cluster, whereas five samples with negative silhouette values in the red boxes were discarded. (**D**,**E**) Kaplan–Meier curves showing time to progression or recurrence in NMIBC patients in the CBNU cohort. *p*-values were calculated using log-rank tests.

**Figure 2 ijms-22-01450-f002:**
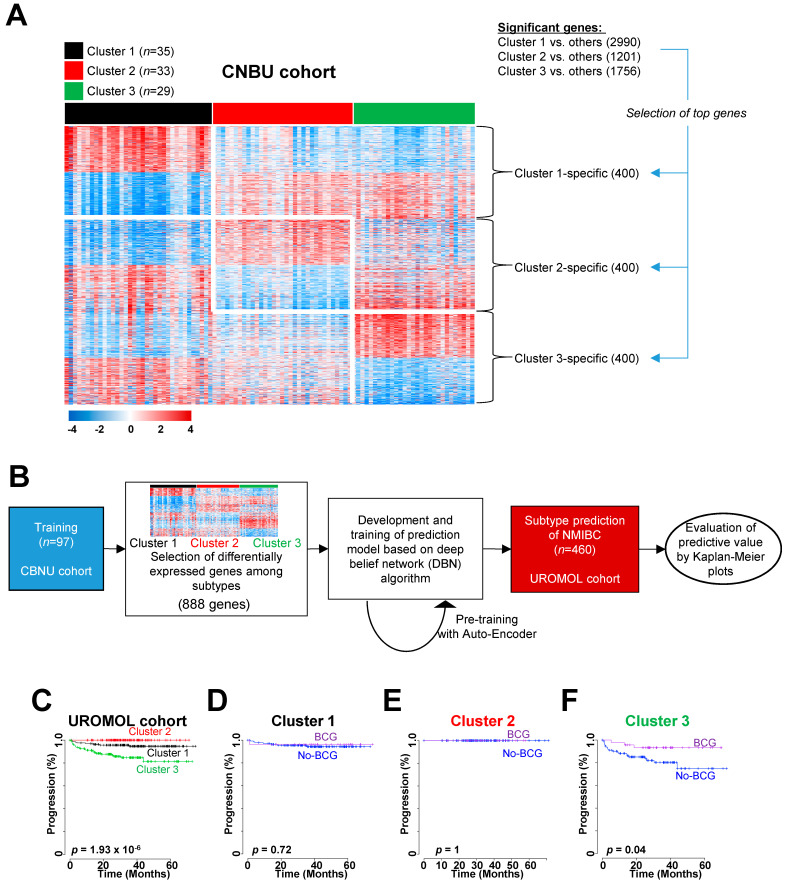
Construction of the subtype classifier and validation of its prognostic value in the UROMOL cohort. (**A**) Expression pattern of cluster-specific genes. Data are presented in matrix format, in which rows represent individual genes and columns represent each tissue. Red and blue reflect high and low expression levels, respectively. (**B**) Validation strategy used to construct a prediction model, and evaluation of predicted outcomes based on 888 genes. (**C**) Kaplan–Meier plots of progression of non-muscle-invasive bladder cancer (NMIBC) patients from the UROMOL cohort, as predicted by a deep belief network algorithm. (**D**–**F**) Ability of the 888 gene-based classifier to predict a response to BCG treatment in the three subgroups. Patients in Cluster 3 (**F**) derived significant benefit from BCG treatment. Data were plotted according to whether patients received BCG therapy. CBNU, Chungbuk National University Hospital; BCG, Bacillus Calmette-Guérin.

**Figure 3 ijms-22-01450-f003:**
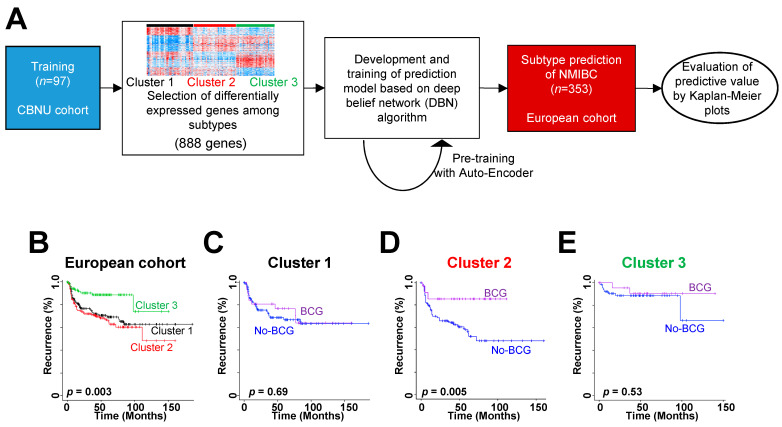
Construction of the subtype classifier and validation of its prognostic value in the European cohort. (**A**) Validation strategy used to construct a prediction model, and evaluation of predicted outcomes based on 888 genes. (**B**) Kaplan–Meier plots of recurrence in non-muscle-invasive bladder cancer (NMIBC) patients from the European cohort, as predicted by a deep belief network algorithm. (**C**–**E**) Prediction of response to BCG treatment in the three subgroups using the 888 gene-based classifier. Patients in Cluster 2 (**D**) derived significant benefit from BCG treatment. Data were plotted according to whether the patients received BCG therapy. CBNU, Chungbuk National University Hospital; BCG, Bacillus Calmette-Guérin.

**Figure 4 ijms-22-01450-f004:**
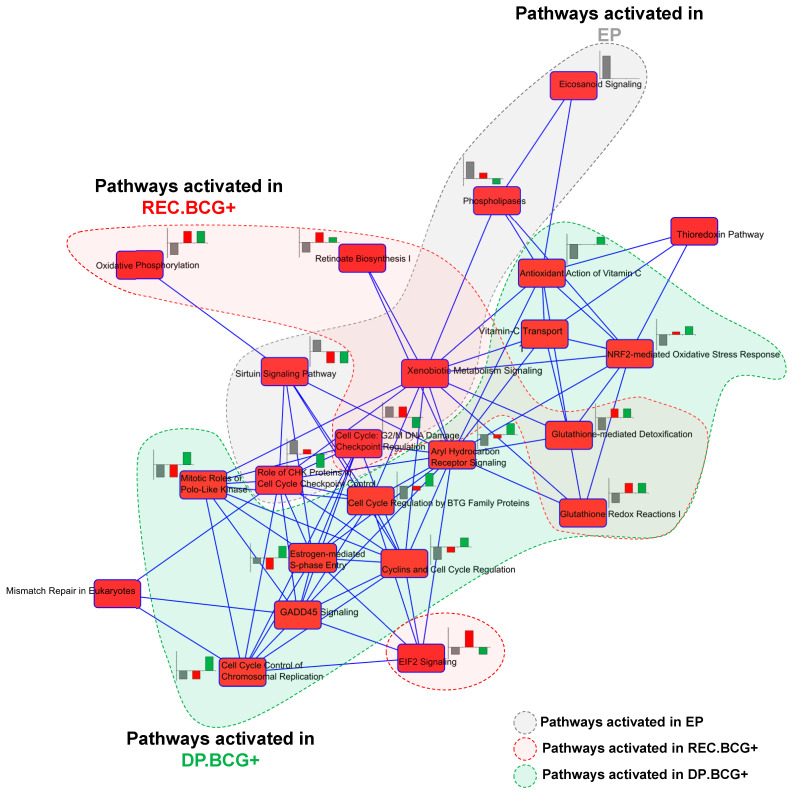
Relationship between canonical pathways significantly associated with molecular subtype. Nodes indicate significant pathways and lines indicate interconnections between two pathways, if they shared molecules. The bar chart beside each pathway node shows activated *z*-scores for the three subtypes: gray, red, or green bars correspond to the activated *z*-score for EP, REC.BCG+, and DP.BCG+, respectively.

**Figure 5 ijms-22-01450-f005:**
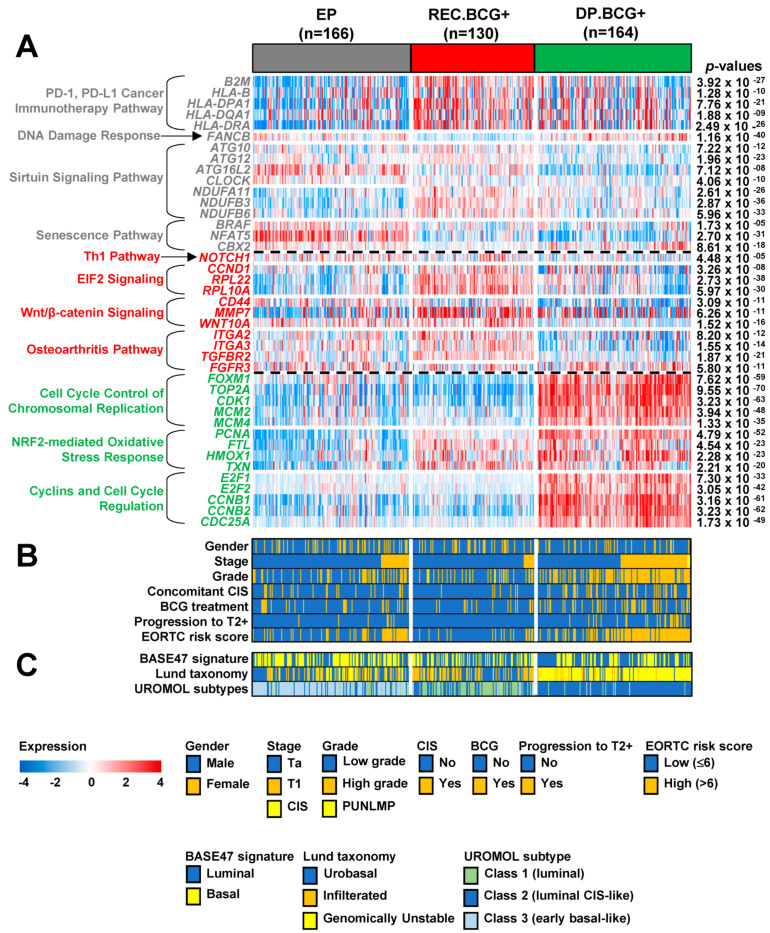
Association between the three molecular subtypes classified by the MSP888 predictor and core molecular features of non-muscle-invasive bladder cancer (NMIBC). The molecular characteristics of the MSP888 subgroups were categorized using heat maps showing expression of genes involved in core pathways (**A**), clinicopathological factors (**B**), and known molecular subtypes (**C**). The *p*-values for the gene expression categories were calculated using one-way analysis of variance (ANOVA). Abbreviations: CIS, Carcinoma in situ; BCG, Bacillus Calmette-Guérin; EORTC, European Organization for Research and Treatment of Cancer; MSP888, molecular subtype prediction of 888 genes.

**Table 1 ijms-22-01450-t001:** Univariate and multivariate Cox regression analysis in the validation cohorts.

Variable		Univariate		Multivariate	
		HR (95% CI)	*p*-Value	HR (95% CI)	*p*-Value
**Disease progression in the UROMOL cohort**	Gender				
	male (Ref.) vs. female	1.219 (0.545–2.725)	0.63		
	Age				
	≤65 (Ref.) vs. >65	1.946 (0.838–4.52)	0.121		
	Stage				
	Ta	Ref.		Ref.	
	T1	9.138 (4.202–19.873)	<0.0001 ^#^	7.437 (3.286–16.831)	<0.0001 ^#^
	CIS	0.000 (0.000–4.47E + 276)	0.978	0.000 (0.000–2.121E + 298)	0.979
	Grade				
	PUNLMP	Ref.			
	Low	1829.893 (0.000–8.723E + 83)	0.937		
	High	7927.856 (0.000–3.777E + 84)	0.925		
	Tumor size				
	<3 cm	Ref.			
	≥3 cm	1.723 (0.754–3.936)	0.197		
	unknown	0.599 (0.204–1.760)	0.351		
	Subtypes *				
	Clusters 1 and 2 (Ref.) vs. 3 *	5.899 (2.635–13.204)	<0.0001 ^#^	3.447 (1.483–8.012)	0.004 ^#^
**Disease recurrence in the European cohort**	Gender				
	male (Ref.) vs. female	1.063 (0.594–1.901)	0.838		
	Age				
	≤65 (Ref.) vs. > 65	0.904 (0.571–1.431)	0.666		
	Stage				
	Ta	Ref.		Ref.	
	T1	24.372 (7.673–77.417)	<0.0001 ^#^	21.090 (6.613–67.262)	<0.0001 ^#^
	CIS	36.658 (6.103–220.177)	<0.0001 ^#^	69.542 (10.365–466.567)	<0.0001 ^#^
	Grade				
	PUNLMP	Ref.			
	Low	0.371 (0.183–0.752)	0.006 ^#^		
	High	1.390 (0.709–2.724)	0.338		
	Subtypes *				
	Clusters 1 and 2 (Ref.) vs. 3 **	3.479 (1.597–7.577)	0.002 ^#^	2.569 (1.065–6.915)	0.036 ^#^

* Predicted outcome in Figure 2C was used for analysis. ** Predicted outcome in Figure 3B was used for analysis. Clusters 1, 2, and 3 correspond to EP, REC.BCG+, or DP.BCG+ subtypes, respectively. # *p* < 0.05. Abbreviations: HR, hazard ratio; CI, confidence interval; Ref., reference; PUNLMP, papillary urothelial neoplasm of low malignant potential; EP, equivocal prognosis; REC.BCG+, subtype associated with recurrence and responsive to BCG; DP.BCG+, subtype associated with disease progression and responsive to BCG.

## Data Availability

The gene expression datasets from the CBNU, European, and CBNU-RNAseq cohorts are available in the NCBI GEO public database (https://www.ncbi.nlm.nih.gov/geo/) under data series accession numbers GSE13507, GSE5479, and GSE163899. The gene expression dataset from the European UROMOL consortium is available in the Array Express database (https://www.ebi.ac.uk/arrayexpress/) under accession number E-MTAB-4321.

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
