# Peer review of "A Molecular Signature Determines the Prognostic and Therapeutic Subtype of Non-Muscle-Invasive Bladder Cancer Responsive to Intravesical Bacillus Calmette-Guérin Therapy"

_ijms, 2021, doi:10.3390/ijms22031450_

Round 1
Reviewer 1 Report
In this study, Kim et al. aimed to identify a molecular signature for NMIBC which might be o prognostic value and predictive of response to BCG therapy. Through bioinformatics analyses of 3 independent cohorts, a signature-based subtype predictor - MSP888 – and three distinct molecular subtypes were disclosed. Multivariate analyses confirmed independent prognostic value of MSP888 for NMIBC prognosis. Associations between this classifier and published molecular subtypes (Lund, UROMOL) were also disclosed. Interestingly, MSP888 further predicted biological activity or responsiveness to BCG treatment.
Overall, the manuscript is well-written and organized. Nonetheless, some issues should be clarified:
- In the introduction section, “NMIBC is a histological type” should be corrected, since the terminology NMBIC and MIBC refers to the extension of the disease and not to the histological type.
- Please elaborate on molecular subtypes for NMIBC in the introduction section.
- Regarding the CBNU-RNAseq, the sample type, i.e., FFPE or fresh tissue, should be specified in the Material and Methods section.
- Lines 160-162 should be clarified because Cluster 3 is associated with a good response to BCG and a decrease in disease progression and this is not clear in the main text. The same should be applied when Cluster 2 is discussed.
- Cohort CBNU-RNAseq seems to significantly differ from the remainder (e.g., female predominance) which may impact the results. This should be discussed.
- In the section Material and Methods, 4.7. Other statistical analysis, it should be shown if the cohorts used in this work present a normal distribution enabling the use parametric tests for statistical analysis. If not, statistical analysis should be reformulated using non-parametric tests.
- The authors may attempt to combine MSP888 and EORTC risk group for predicting progression and recurrence in BCG-treated patients, as it may provide better accuracy.
Minor issues
Line 273 – “Comparison”.
Author Response
Reply to the comments of Reviewer 1:
Reviewer #1:
In this study, Kim et al. aimed to identify a molecular signature for NMIBC which might be o prognostic value and predictive of response to BCG therapy. Through bioinformatics analyses of 3 independent cohorts, a signature-based subtype predictor - MSP888 – and three distinct molecular subtypes were disclosed. Multivariate analyses confirmed independent prognostic value of MSP888 for NMIBC prognosis. Associations between this classifier and published molecular subtypes (Lund, UROMOL) were also disclosed. Interestingly, MSP888 further predicted biological activity or responsiveness to BCG treatment.
Overall, the manuscript is well-written and organized. Nonetheless, some issues should be clarified:
- In the introduction section, “NMIBC is a histological type” should be corrected, since the terminology NMBIC and MIBC refers to the extension of the disease and not to the histological type.
Answer: First, we really appreciate the reviewer’s valuable comments. The reviewer’s point is correct. According to the comment, we modified the statements in the Introduction section removing the word ‘histological’.
- Please elaborate on molecular subtypes for NMIBC in the introduction section.
Answer: The reviewer’s comment is informative. To provide more information, we additionally described previously established molecular subtype classification systems in NMIBC including the Lund Taxonomy (Clin Cancer Res. 2012 Jun 15;18(12):3377-86), the UROMOL classes (Cancer Cell. 2016 Jul 11;30(1):27-42), and the five consensus subtypes of T1 tumors (Eur Urol. 2020 Oct;78(4):533-537).
- Regarding the CBNU-RNAseq, the sample type, i.e., FFPE or fresh tissue, should be specified in the Material and Methods section.
Answer: All the samples in the CNBU-RNAseq cohort were fresh frozen tissues. We additionally described it in the subsection 4.2. Patients and tissue samples of a validation cohort of the Materials and Methods section.
- Lines 160-162 should be clarified because Cluster 3 is associated with a good response to BCG and a decrease in disease progression and this is not clear in the main text. The same should be applied when Cluster 2 is discussed.
Answer: Thanks for the comment for clarity. To clarify the meaning of the Cluster 3 (DP.BCG+), we modified the main text from “indicating a subtype associated with disease progression and a good response to BCG treatment” into “indicating a subtype associated with a poor prognosis in disease progression and a good response to BCG treatment” at the last paragraph in the subsection 2.2. One subtype, Cluster 3, is predictive of disease progression and a better response to BCG treatment of the Results section.
Regarding the Cluster 2 (REC.BCG+), we also modified the main text from “indicating a subtype associated with a poor prognosis in disease recurrence and a better response to BCG treatment” into “indicating a subtype associated with a poor prognosis in disease recurrence and a better response to BCG treatment” at the last paragraph in the subsection 2.3. One subtype, Cluster 2, is associated with disease recurrence and a better response to BCG treatment of the Results section.
- Cohort CBNU-RNAseq seems to significantly differ from the remainder (e.g., female predominance) which may impact the results. This should be discussed.
Answer: Sorry to our mistake. Male to female ratio is 26 (81.3) & 6 (18.8). The table was fixed.
- In the section Material and Methods, 4.7. Other statistical analysis, it should be shown if the cohorts used in this work present a normal distribution enabling the use parametric tests for statistical analysis. If not, statistical analysis should be reformulated using non-parametric tests.
Answer: We really appreciate the reviewer’s informative comments. By applying Lilliefors tests, which test the null hypothesis that data come from a normally distributed population, we estimated normality of gene expression data from NMIBC patient cohorts. Notably, all gene expression data used in this study were log2-transformed and quantile-normalized. When we estimated p-values across all genes in the data sets, we could not observed any statistical significance, indicating that the expression data sets showed a normal distribution. Therefore, we stay the results of statistical tests based on two-sample t-test or one-way ANOVA tests. To provide distribution of gene expression data used in this study, we additionally described normality assessments of the data in Figure S4 and in the subsection 4.7. Other statistical analysis of the Materials and Methods section.
- The authors may attempt to combine MSP888 and EORTC risk group for predicting progression and recurrence in BCG-treated patients, as it may provide better accuracy.
Answer: We are really appreciating your valuable comment, and fusion of our molecular subtyping and EORTC risk stratification is the last goal of our study. However, we was wonder MSP888 subtyping would be successfully separate the patient who have high risk of NMIBC those could not be classified by EORTC risk stratification. Thus CBNU-RNAseq cohort (all patients have high risk according to ERORC stratification) was tested as validation of MSP888, and the molecular subtyping was successfully separating the patients as their clinical outcomes. Our next step would be the combination of MSP888 & EORTC risk group with unsupervised consecutive NMIBC patients.
Minor issues
Line 273 – “Comparison”.
Answer: As the reviewer pointed out, we fixed the word. Thanks again for your valuable review.
Reviewer 2 Report
This paper showed the new molecular signature and its role in predicting the response to BCG therapy in NMIBC. Several points should be revised prior to publication.
Major points
1) In the CBNU cohort, 83.5% of the patients had low-grade tumors (Table S1). Nevertheless, 77.7% of the patients had T1 tumors. The accuracy and reliability of pathological diagnosis should be confirmed.
2) In Table 1, the reference group of each variable is unclear, especially in stage, grade, and subtype. Only one HR was described in each variable, but if the three groups were analyzed in stage, grade, and subtypes, I suppose two HRs should be described in the Table.
3) According to the results, the EP subtype is considered as refractory to BCG therapy. Is there any data regarding FGFR expression in this group. Some previous studies reported that tumors with high FGFR expression was associated with poor response to BCG therapy.
4) Based on the results of the pathway analysis, what treatment strategy do the authors propose for the patients with the EP subtype?
Minor points
1) In line 273, "Comaprison" should be corrected.
2) It is better to describe the definitions of recurrence and progression. Maybe they might be a bit different between the cohorts.
Author Response
Reply to the comments of Reviewer 2:
Reviewer #2:
This paper showed the new molecular signature and its role in predicting the response to BCG therapy in NMIBC. Several points should be revised prior to publication.
Major points
1) In the CBNU cohort, 83.5% of the patients had low-grade tumors (Table S1). Nevertheless, 77.7% of the patients had T1 tumors. The accuracy and reliability of pathological diagnosis should be confirmed.
Answer: Sorry to our mistake in CBNU cohort. CBNU cohort consisted with old patients who performed operation from 1996 to 2007. Thus most of patients have WHO grading system of 1973. We modify the table S1 as below.
Table S1. Baseline characteristics of NMIBC patient cohorts.
|
Variables |
CBNU cohort |
UROMOL cohort |
European cohort |
CBNU-RNAseq cohort |
|
Patients (n) |
103 |
460 |
353 |
32 |
|
Gender, n (%) |
||||
|
Male |
87 (84.5) |
357 (77.6) |
282 (79.9) |
26 (81.3) |
|
Female |
16 (15.5) |
103 (22.4) |
68 (19.3) |
6 (18.8) |
|
NA |
3 (0.8) |
|||
|
Age (years) |
||||
|
Median |
66 |
69 |
69 |
72 |
|
Range |
24-88 |
23-96 |
27-95 |
24–81 |
|
Stage, n (%) |
||||
|
Ta |
23 (22.3) |
345 (75) |
188 (53.3) |
10 (31.3) |
|
T1 |
80 (77.7) |
112 (24.3) |
162 (45.9) |
22 (68.8) |
|
Other |
3 (0.7) |
3 (0.8) |
||
|
Grade (1973 system), n (%) |
|
|
|
|
|
Grade 1 |
29 (28.2) |
|
|
|
|
Grade 2 |
57 (55.3) |
|
|
|
|
Grade 3 |
17 (16.5) |
|
|
|
|
Grade (2004 system), n (%) |
||||
|
PUNLMP |
|
7 (1.5) |
33 (9.3) |
|
|
Low |
|
277 (60.2) |
97 (27.5) |
21 (65.6) |
|
High |
|
176 (38.3) |
223 (63.2) |
11 (34.4) |
|
CIS |
||||
|
No |
103 (100) |
389 (84.6) |
302 (85.6) |
32 (100) |
|
Yes |
71 (15.4) |
51 (14.4) |
||
|
Intravesical Therapy, n (%) |
||||
|
No |
47 (45.6) |
372 (80.9) |
263 (74.5) |
2 (6.3) |
|
Yes |
56 (54.4) |
88 (19.1) |
90 (25.5) |
30 (93.8) |
|
Median follow-up (months) |
71.3 |
33 |
52 |
75.4 |
|
Abbreviations: NMIBC, non-muscle invasive bladder cancer; CBNU, Chungbuk National University Hospital; NA, not available |
||||
2) In Table 1, the reference group of each variable is unclear, especially in stage, grade, and subtype. Only one HR was described in each variable, but if the three groups were analyzed in stage, grade, and subtypes, I suppose two HRs should be described in the Table.
Answer: Thanks for your valuable comment. We reevaluate the uni- and multi-variate analysis as your comment. The table are below. In the first analysis, we realized that some of well-known prognostic values did not show the statistically significant in Cox regression analysis in UROMOL & European cohorts. For example, grade of European cohort did not show the significant for recurrence. Low grade had protective effect of recurrence compare to PUNLMP grade. We also don’t completely understand why it was happened, but these public data would be collected from multiple hospitals with heterogenous clinical information. So, these factors would be bias of these data.
Table 1. Univariate and multivariate Cox regression analysis in the validation cohorts.
|
Variable |
Univariate |
Multivariate |
|||
|
HR (95% CI) |
p-value |
HR (95% CI) |
p-value |
||
|
Disease progression in the UROMOL cohort |
Gender |
||||
|
male (Ref.) vs. female |
1.219 (0.545 - 2.725) |
0.63 |
|
|
|
|
Age |
|||||
|
≤ 65 (Ref.) vs. > 65 |
1.946 (0.838 - 4.52) |
0.121 |
|
|
|
|
Stage |
|||||
|
Ta |
Ref. |
|
Ref. |
|
|
|
T1 |
9.138 (4.202-19.873) |
<0.0001# |
7.437 (3.286-16.831) |
<0.0001# |
|
|
CIS |
0.000 (0.000-4.47E+276) |
0.978 |
0.000 (0.000-2.121E+298) |
0.979 |
|
|
Grade |
|||||
|
PUNLMP |
Ref. |
|
|
|
|
|
Low |
1829.893 (0.000-8.723E+83) |
0.937 |
|
|
|
|
High |
7927.856 (0.000-3.777E+84) |
0.925 |
|
|
|
|
Tumor size |
|||||
|
<3cm |
Ref. |
|
|
|
|
|
≥3cm |
1.723 (0.754-3.936) |
0.197 |
|
|
|
|
unknown |
0.599 (0.204-1.760) |
0.351 |
|
|
|
|
Subtypes* |
|||||
|
Cluster 1&2 (Ref.) vs. 3 |
5.899 (2.635-13.204) |
<0.0001# |
3.447 (1.483 -8.012) |
0.004# |
|
|
Disease recurrence in the European cohort |
Gender |
||||
|
male (Ref.) vs. female |
1.063 (0.594 - 1.901) |
0.838 |
|
|
|
|
Age |
|||||
|
≤ 65 (Ref.) vs. > 65 |
0.904 (0.571-1.431) |
0.666 |
|
|
|
|
Stage |
|||||
|
Ta |
Ref. |
|
Ref. |
|
|
|
T1 |
24.372 (7.673-77.417) |
<0.0001# |
21.090 (6.613-67.262) |
<0.0001# |
|
|
CIS |
36.658 (6.103-220.177) |
<0.0001# |
69.542 (10.365-466.567) |
<0.0001# |
|
|
Grade |
|
|
|
|
|
|
PUNLMP |
Ref. |
|
|
|
|
|
Low |
0.371 (0.183-0.752) |
0.006# |
|
|
|
|
High |
1.390 (0.709 -2.724) |
0.338 |
|
|
|
|
Subtypes* |
|
|
|
|
|
|
Cluster 1&2 (Ref.) vs. 3 |
3.479 (1.597 – 7.577) |
0.002# |
2.569 (1.065-6.915) |
0.036# |
|
* Predicted outcome in Figure 2C was used for analysis. ** Predicted outcome in Figure 3B was used for analysis. Clusters 1, 2, and 3 correspond to EP, REC.BCG+, or DP.BCG+ subtypes, respectively. #p< 0.05. Abbreviations: HR, hazard ratio; CI, confidence interval; Ref., reference.
3) According to the results, the EP subtype is considered as refractory to BCG therapy. Is there any data regarding FGFR expression in this group. Some previous studies reported that tumors with high FGFR expression was associated with poor response to BCG therapy.
Answer: The reviewer’s comment is very informative. According to the query of reviewer, we compared expression levels of FGFR3 between EP and non-EP subgroups. Across patient cohorts except for the CBNU-RNAseq cohort containing small samples, FGFR3 expression levels in the EP subtype were significantly higher than in the non-EP subgroups (Figure S2). These results support a poor responsiveness to BCG treatment in the EP subtype and are consistent with the previous investigation (J Urol. 2016 Jan;195(1):188-97). To provide information an association between EP subtype and FGFR3 expression, we additionally described these results in the subsection 2.5. Biologic characteristics of the molecular subtypes of NMIBC of the Result section along with Figure S2.
4) Based on the results of the pathway analysis, what treatment strategy do the authors propose for the patients with the EP subtype?
Answer: As we already described in the Results and Discussion sections, PD-1/PD-L1 cancer immunotherapy and DNA damage responses are actively associated with the EP subtype. These results imply that PD-1/PD-L1 inhibitors may be a suitable treatment for NMIBC patients with the EP subtype. However, the effectiveness of PD-1/PD-L1 inhibitors against the EP subtype should be validated in other NMIBC cohorts since there is no treatment data relevant to PD-1/PD-L1 inhibitors in patients with NMIBC.
Minor points
1) In line 273, "Comaprison" should be corrected.
Answer: As the reviewer pointed out, we fixed the word.
2) It is better to describe the definitions of recurrence and progression. Maybe they might be a bit different between the cohorts.
Answer: As described in the subsection 4.2. Patients and tissue samples of a validation cohort of the Material and Methods section, disease recurrence was defined as relapse of primary NMIBC at a lower or equivalent pathologic stage (Ta or T1), whereas disease progression was defined as an increase in stage from either Ta or T1 to T2 or higher after relapse. For more information, we also cited the literature that described disease recurrence and progression in NMIBC. Thanks again for the reviewer’s valuable comments.
Round 2
Reviewer 1 Report
The authors have adequately addressed all the issues raised by this reviewer.